# Role of gender in perspectives of discrimination, stigma, and attitudes relative to cervical cancer in rural Sénégal

**Natalia Ongtengco**[1]☯, **Hamidou Thiam**[2]☯, **Zola Collins**[3]☯, **Elly Lou De Jesus**[3]☯, **Caryn E. Peterson**[3,4,5]☯, **Tianxiu Wang**[3]☯, **Ellen Hendrix**[3]☯, **Youssoupha Ndiaye**[6]‡, **Babacar Gueye**[7]‡, **Omar Gassama**[8]‡, **Abdoul Aziz Kasse**[9]‡, **Adama Faye**[10]‡, **Jennifer S. Smith**[11]‡, **Marian Fitzgibbon**[3,5]‡, **Jon Andrew Dykens**[3,5,12,13]☯ *

**1** University of Illinois at Chicago, Chicago, Illinois, United States of America, **2** Office of Training, Supervision, and Research, Kédougou Medical Region, Kédougou, Sénégal, **3** Institute for Health Research and Policy, University of Illinois at Chicago, Chicago, Illinois, United States of America, **4** Division of Epidemiology and Biostatistics, School of Public Health, University of Illinois at Chicago, Chicago, Illinois, United States of America, **5** Cancer Center, University of Illinois at Chicago, Chicago, Illinois, United States of America, **6** Division of Planning, Research and Statistics, Sénégal Ministry of Health and Social Action, Dakar, Sénégal, **7** Division of Noncommunicable Diseases, Center for Disease Control, Sénégal Ministry of Health and Social Action, Dakar, Sénégal, **8** Service of Obstetrics and Gynecology, University of Cheikh Anta Diop, Dakar, Sénégal, **9** Cancer Institute, University of Cheikh Anta Diop, Dakar, Sénégal, **10** Institute of Health and Development, University of Cheikh Anta Diop, Dakar, Sénégal, **11** School of Public Health, University of North Carolina, Chapel Hill, North Carolina, United States of America, **12** Center for Global Health, College of Medicine, University of Illinois at Chicago, Chicago, Illinois, United States of America, **13** Department of Family Medicine, College of Medicine, University of Illinois at Chicago, Chicago, Illinois, United States of America

☯ These authors contributed equally to this work.
‡ These authors also contributed equally to this work.
* jdykens@uic.edu

**Data Availability Statement:** All relevant data are within the paper and its Supporting Information files.

## Abstract

Cervical cancer is the leading cause of female cancer deaths in Sénégal which is ranked 17[th] in incidence globally, however, the screening rate there is very low. Nuanced gendered perceptions and health behaviors of both women and men play a significant role in women's health. Our study analyzed gender differences on perceptions of gender roles, discrimination, cancer attitudes, cancer stigma, and influences in healthcare decision making within our study population to inform ongoing cervical cancer prevention work in the rural region of Kédougou, Sénégal. We conducted a cross-sectional survey of 158 participants, 101 women and 57 men (ages 30–59) across nine non-probability-sampled communities from October 2018 through February 2019. Bivariate analysis was conducted to assess gender differences across all variables. We also conducted analyses to determine whether there were significant differences in beliefs and attitudes, by screening behavior and by education. We found significant gender differences regarding the perception of a woman's role (P < 0.001) and a man's role (P = 0.007) as well as in the everyday discrimination questions of "decreased respect by spouse" (P < 0.001). Regarding cancer stigma, among women, 18.00% disagreed and 10.00% strongly disagreed while among men, 3.6% disagreed and 1.8% strongly disagreed that "If I had cancer, I would want my family to know that I have it." When making decisions about one's healthcare, women are more likely than men to trust

**Funding:** Research reported in this publication was supported by the Fogarty International Center of the National Institutes of Health under Award Number K01TW010494 to JAD. The content is solely the responsibility of the authors and does not necessarily represent the official views of the National Institutes of Health.

**Competing interests:** The authors have declared that no competing interests exist.

social contacts such as their spouse (46.5% vs 5.3%, p < 0.001) while men are more likely than women to trust health service personnel such as a nurse (50.9% vs 18.8%, p < 0.001). Furthermore, men and women were both more likely to state that men have the final decision regarding the healthcare decisions of women (p < 0.001). Our data reveal structural disadvantages for women within our study population as well as gender differences in the adapted everyday discrimination scale and cancer stigma scale. Higher rates of both personal and perceived cancer stigma among women has profound implications for how population and community level communication strategies for cancer prevention and control should be designed. Efforts to advance the goal of the elimination of cervical cancer should, in the short-term, seek to gain a more profound understanding of the ways that gender, language, and other social determinants impact negative social influences and other barriers addressable through interventions. Social and behavior change communication may be one approach that can focus both on education while seeking to leverage the social influences that exist in achieving immediate and long-term goals.

## Introduction

There are over half a million cervical cancer cases diagnosed annually making it the third most common cancer in women worldwide. [1] Additionally, it is the leading cause of female cancer deaths in Sénégal with an estimated 1,876 cervical cancer cases diagnosed annually with 1,367 deaths resulting in a age-standardized mortality rate of 29.1 compared to 6.9 globally. [2] The age-standardized cervical cancer incidence rate in North America is 7.6/100,00 women compared to 23/100,000 in Western Africa and 37.8/100,000 in Sénégal, ranking it the 17th highest incidence in the world. [3,4] Despite the effectiveness of cervical cancer screening and treatment in reducing incidence and mortality, [5] the estimated participation rate for cervical cancer screening in Sénégal is very low (6.9% of all women ages 18 to 69). It is especially low in rural areas and in older age groups (1.9% of women ages 40 to 49 and 0% for women 50 and above). [6] Cervical cancer screening remains unavailable in many rural areas of Senegal but has been accessible throughout the Kédougou region of Sénégal since 2014 through the efforts of an ongoing partnership. [7] Cervical cancer is both preventable and concentrated in low- and middle-income countries (LMICs) [8] with over 85% of global cervical cancer deaths occurring in LMICs. [3] The high incidence and mortality of cervical cancer is an important indicator of larger health system problems, including poor access to care and screening and the lack of culturally competent communication; factors that disproportionately affect poor women. [9]

Gender is recognized as an important social determinant of health. [10] In many contexts there is a structural disadvantage for women that goes beyond the fact of the illnesses affecting them. Nuanced gendered perceptions and health behaviors of both women and men play a significant role in women's ability to access the care that they need. [11] Men or older family members are often the decision makers for when and how women may gain access to healthcare. [12] In addition, when women are empowered, their increased decision-making autonomy and access to economic resources have a positive effect on their use of healthcare services. [13,14] Conversely, perceived discrimination reduces the likelihood of seeking cervical cancer screenings, [15] and shame and stigma limits women's overall use of health services. [12]

Previous studies have found important gender differences for stigmatized illnesses such as HIV/AIDS and mental illness in LMICs. [16–22] In Kenya and Pakistan, women had higher personal stigma attitudes than men toward HIV/AIDS, depression, and HPV infection. [18,21,23] There are also gender differences for acceptability of treatment, including seeking psychological help and getting vaccinated. [16,24,25] Many of these studies emphasize the need to incorporate gender-specific components in interventions to increase acceptability and healthcare utilization. [16,17,21,24,26] For example, a study found male-female differences on effective strategies to increase HPV vaccine acceptability: for men, it was most effective to correct misconceptions, promote healthcare provider recommendations, and emphasize perceived benefits, whereas for women, it was more important to address gender norms and discrimination. [24] A meta-analysis on gender differences related to HPV vaccine acceptability found similar results. [17]

There is limited research on stigma associated with cervical cancer. Social stigmas around sexual behavior and HPV infection [27] may contribute to vaccine and screening hesitancy. Social stigma can manifest as personal stigmas (i.e., how one views and treats others) or as perceived public stigmas (i.e., how one thinks others view and treat them). [28] In addition, the expression of attitudes related to stigma is moderated by social influence—that is by the ability of individuals to affect one another's thoughts, ideas, and behaviors. [29–31] In these ways, negative social influences play a role in spreading negative behaviors [29,32] and may be linked to cervical cancer screening hesitancy. Understanding how these factors contribute to the acceptability and adoption of cervical cancer prevention is of paramount importance. A broader knowledge of the relationship of gender differences within stigma, discrimination, and acceptability may help to improve the global response to cervical cancer. Our study analyzed gender differences on perceptions of gender roles, discrimination, cancer attitudes, cancer stigma, and influences in healthcare decision making within our study population.

## Methods

We conducted a cross-sectional survey of 158 participants, 101 women and 57 men (ages 30–59) across nine non-probability-sampled communities (two rural and one semi-urban from each district, across three districts) in the Kédougou region of Sénégal from October 2018 through February 2019. We collected demographic information and data on health service utilization, cervical cancer knowledge, and experience of cervical cancer screening through interviewer-administered surveys. The surveys were administered to one woman and one man per 10 randomly selected households and five women per women's group within each community. Survey interviews were conducted in the participants' choice of language: French, Malinke, or Pulaar. Participants were eligible for inclusion in the study if they were between 30–59 years old and were female or were a male living with a female who is able to seek cervical cancer prevention services from a health facility in the Kédougou region of Sénégal. Participants who were outside the target age-range were not eligible for participation.

### Site selection and recruitment

The region of Kédougou is divided into three medical districts: Kédougou, Saraya, and Salemata. Each of these health districts has a single health center in the district capital and multiple health posts in the rural surrounding communities. We selected nine sites in the Kédougou Medical Region through non-probability sampling including one health center and two rural health posts from each of the three districts comprising the region. In the Kédougou District we selected the Dalaba health post (population accessing this health post = 5995), Bandafassi (7189), and Dindefello (9370). In the Salemata District we selected the Salemata Health Center

(7278) and the health posts of Dar Salaam (3084) and Dakately (3037). In the Saraya District, we selected the Saraya Health Center (5890) and the posts of Nafadji (3759) and Khossanto (3471). Each of the nine sites was mapped through OpenStreetMaps. [33] Printable maps were created (using FieldPapers [34]) to illustrate structures (assumed to be households), roads, and rivers in each site. Maps were divided into four sectors with approximately the same number of structures in each quadrant. Structures were numbered and Google's random number generator was used to determine the starting point. Counting in increments of the limiting factor, each chosen structure was marked and recorded. This ensured a relatively even distribution of structures selected throughout each site. Twenty structures per site (n = 180) were selected and visited in order to assess for eligibility. Potential participants were recruited using the approved recruitment script. Households were selected if there was both an eligible woman and man who agreed to participate. An additional five women were recruited in each site from among the women's group to strengthen the assessment at the community level. The target sample size of 225 (135 women and 90 men) was determined based on the need to have a heterogeneous sample across language and district. Our sample was not adequately powered to detect differences between screened and unscreened women.

## Development of documents

The questionnaire included closed-ended, quantitative questions seeking information on participant's perceptions of discrimination, cancer stigma, opinions, and attitudes. We included adapted questions from the Everyday Discrimination Scale [35] and the Cancer Stigma Scale. [36] The questionnaires were first created in English, translated into French and the local languages of Jakhanke/Malinke and Pula Fuuta (a dialect of Fula/Pulaar), and then back-translated for accuracy by certified Sénégalese translators. Questionnaires were field tested for comprehension prior to the initiation of the study. All IRB approved documents including study overview, recruitment scripts, and the informed consent were available in French, Malinke, and Pulaar.

## Consent and data collection

All research assistant data collectors participated in a three-day training on the project protocol including data collection methodology facilitated by the lead investigator prior to field testing the instrument. After final institutional review board approval, research assistants attended an additional three-day training to review all data collection procedures. The study research assistants read the informed consent aloud, in the participants preferred language, and participants reviewed and signed the approved informed consent short form, written in French. In cases where the participant did not read French, a trusted contact was requested by the participant to witness the informed consent process, observe the signature of the participant, and then sign as a witness. After participants were consented, data collection was conducted immediately with a female research assistant collecting data from women and a male research assistant collecting data from men. All data collection activities were performed in a private setting. Data collection interviews occurred in the preferred language of the participant. All responses were recorded on hard copy interview forms with the name of the participant being the only item recorded on the final page. All data collection instruments were immediately handed over to the lead research assistant, who recorded the participant's name on the participant code book, placed a unique identifier on page one of the data collection instrument, removed and destroyed the final page of each instrument, scanned all documents, and transmitted them through a secure portal to a research assistant in the United States.

## Data analysis

Data were double-entered into an electronic spreadsheet by two research assistants, compared, approved and subsequently cleaned by the principal investigator. Bivariate analyses were conducted to 1) assess gender differences in the distribution of all variables including the adapted everyday discrimination scale, cancer stigma scale, and various cervical cancer attitude measures; 2) determine whether there were significant differences in beliefs and attitudes by screening behavior; and 3) explore the potential effect of educational attainment on gender differences in attitudes and beliefs. To accomplish the third analysis, we created a composite variable combining gender and education, categorizing those with Quranic school or no school as having "Low Education" and those who attended primary school, secondary school, and above as having "Higher Education." Associations were tested using the Fisher's Exact test statistic (we computed the p-value for analyses by way of the Two- Stage Fisher's Exact Test using RStudio version 1.2.1578 through the Dplyr and Arsenal packages).

## The conduct of responsible research and partnership

The University of Illinois at Chicago has an ongoing partnership affiliation agreement with the Institute of Health and Development at the University Cheikh Anta Diop and with the Medical Region of Kédougou. This partnership uses a participatory approach ensuring that all activities are well-aligned with the expressed priorities of the local health system. This human subjects research was approved by the University of Illinois at Chicago Institutional Review Board and the Institutional Review Board at the Ministry of Health and Social Action in Senegal. The Medical Region of Kédougou, the three health districts, and participating health posts granted researchers permission through signed letters of support to implement and conduct the data collection activities. Each investigator and U.S. based research assistant received the Collaborative Institutional Training Initiative (CITI) training certification prior to conducting the research. [37] All local research assistants were trained in research ethics through a locally approved ethics of human research training program.

## Results

Our enrollment goal was 225 (135 women and 90 men). With 158 participants (101 women and 57 men), we achieved a functional response rate of 70.2%. The response rate for women was appreciably higher than for men (74.8% and 63.3%, respectively). The mean age of participants was 41.6 with the mean age of men (44.1) being slightly higher than the mean age of women (40.2). The distribution of participants across sites is reported in Table 1. There were significant gender differences in educational level (P <0.001). Among those surveyed, 97% of all women and 76.8% of all men had no more than a primary education while 25.7% of women and 10.3% of men had no formal education at all. No women in our sample attended more than two years of secondary school while 8.9% of men were educated beyond two years of secondary school and an additional 7.1% had at least some university education. The majority of participants (92.1% of women and 94.7% of men) were married, and among all women, 51.5% were in a polygamous household (P = 0.004). The majority of our sample speaks one or both prevalent local languages, Malinke (62.7%) and Pulaar (59.5%). As is characteristic for the region, there are fewer Wolof (26.6%) and French (31%) speakers in our sample. It should be noted that there are significantly more male Wolof (36.8%) than female Wolof (20.8%) speakers (P = 0.039) as well as male French (45.6%) than female French (22.8%) speakers (P = 0.004). Among the women in our sample, 84.2% have never been screened for cervical cancer, 13.9% have been screened one time, and 2.0% have been screened multiple times. (Table 1)

**Table 1. Demographics by gender.**

| | Female (N = 101) | Male (N = 57) | Total (N = 158) | p value |
|---|---|---|---|---|
| **Age in years** | | | | 0.006 |
| Mean (SD) | 40.168 (8.631) | 44.140 (8.355) | 41.601 (8.718) | |
| Range | 30.000–59.000 | 30.000–59.000 | 30.000–59.000 | |
| **Community** | | | | 0.785 |
| Salemata District—Salemata | 9 (8.9%) | 5 (8.8%) | 14 (8.9%) | |
| Salemata District—Dar Salaam | 5 (5.0%) | 5 (8.8%) | 10 (6.3%) | |
| Salemata District—Dakately | 10 (9.9%) | 9 (15.8%) | 19 (12.0%) | |
| Saraya District—Saraya | 15 (14.9%) | 7 (12.3%) | 22 (13.9%) | |
| Saraya District—Nafadji | 15 (14.9%) | 4 (7.0%) | 19 (12.0%) | |
| Saraya District—Khossanto | 8 (7.9%) | 7 (12.3%) | 15 (9.5%) | |
| Kedougou District—Dalaba | 12 (11.9%) | 6 (10.5%) | 18 (11.4%) | |
| Kedougou District—Bandifassi | 15 (14.9%) | 8 (14.0%) | 23 (14.6%) | |
| Kedougou District—Dindefello | 12 (11.9%) | 6 (10.5%) | 18 (11.4%) | |
| **Education level** | | | | < 0.001 |
| None | 26 (25.7%) | 5 (8.9%) | 31 (19.7%) | |
| Quranic School | 35 (34.7%) | 21 (37.5%) | 56 (35.7%) | |
| Primary education | 37 (36.6%) | 17 (30.4%) | 54 (34.4%) | |
| Secondary school through university | 3 (3.0%) | 13 (23.2%) | 16 (10.2%) | |
| **Marital status** | | | | 0.071 |
| Single, divorced, separated, or widowed | 8 (7.9%) | 3 (5.3%) | 11 (7.0%) | |
| Married (monogamous household) | 41 (40.6%) | 34 (59.6%) | 75 (47.5%) | |
| Married (polygamous household) | 52 (51.5%) | 20 (35.1%) | 72 (45.6%) | |
| **Malinke speaker** | | | | 0.733 |
| No | 39 (38.6%) | 20 (35.1%) | 59 (37.3%) | |
| Yes | 62 (61.4%) | 37 (64.9%) | 99 (62.7%) | |
| **Pulaar speaker** | | | | 0.019 |
| No | 48 (47.5%) | 16 (28.1%) | 64 (40.5%) | |
| Yes | 53 (52.5%) | 41 (71.9%) | 94 (59.5%) | |
| **Wolof speaker** | | | | 0.039 |
| No | 80 (79.2%) | 36 (63.2%) | 116 (73.4%) | |
| Yes | 21 (20.8%) | 21 (36.8%) | 42 (26.6%) | |
| **French speaker** | | | | 0.004 |
| No | 78 (77.2%) | 31 (54.4%) | 109 (69.0%) | |
| Yes | 23 (22.8%) | 26 (45.6%) | 49 (31.0%) | |
| **Screened for cervical cancer** | | | | |
| Never screened | 85 (84.2%) | 0 | 85 (84.2%) | |
| One time only | 14 (13.9%) | 0 | 14 (13.9%) | |
| More than one time | 2 (2.0%) | 0 | 2 (2.0%) | |

We found significant gender differences regarding the perception of a woman's role (P < 0.001) and a man's role (P = 0.007). Among women, 7.0% agreed and 59.0% strongly agreed that a woman's most important role is to take care of her home and cook for her family, while among men, 43.9% agreed and 40.4% strongly agreed with this statement. Concerning a man's role, 14.9% of women agreed and 53.5% strongly agreed that a man should have the final word about decisions in his home, while among men, 23.2% agreed and 69.6% strongly agreed. (Table 2)

**Table 2. Perception of gender roles by gender.**

| | Female (N = 101) | Male (N = 57) | Total (N = 158) | p value |
|---|---|---|---|---|
| **A woman's most important role is to take care of her home and cook for her family** | | | | < 0.001 |
| Strongly Disagree | 6 (6.0%) | 1 (1.8%) | 7 (4.5%) | |
| Disagree | 26 (26.0%) | 6 (10.5%) | 32 (20.4%) | |
| Undecided | 2 (2.0%) | 2 (3.5%) | 4 (2.5%) | |
| Agree | 7 (7.0%) | 25 (43.9%) | 32 (20.4%) | |
| Strongly Agree | 59 (59.0%) | 23 (40.4%) | 82 (52.2%) | |
| **A man should have the final word about decisions in the home** | | | | 0.009 |
| Strongly Disagree | 7 (6.9%) | 1 (1.8%) | 8 (5.1%) | |
| Disagree | 24 (23.8%) | 3 (5.4%) | 27 (17.2%) | |
| Undecided | 1 (1.0%) | 0 (0.0%) | 1 (0.6%) | |
| Agree | 15 (14.9%) | 13 (23.2%) | 28 (17.8%) | |
| Strongly Agree | 54 (53.5%) | 39 (69.6%) | 93 (59.2%) | |

Within our study population, we found significant gender differences in the everyday discrimination questions of "decreased respect by spouse" (P < 0.001) and "others act toward them as if they are not smart" (P = 0.031). Specifically, 48.0% of women and 78.8% of men stated that they never experienced disrespect by their spouse, while, in contrast, 14.3% of women and 0% of men stated that they are treated with less courtesy or respect by their spouse on a daily basis. Men were significantly more likely to feel that others acted as if they are not smart. Of the men who stated that they perceived this type of discrimination, 32.1% stated that this occurred a few times in their life, 9.4% that it occured a few times per year and none stated that it occured weekly or daily. Within our study, 78.2% of women and 58.5% of men stated that they were never perceived as unintelligent by others. Of the women who stated that they perceived this type of discrimination, 15.8% stated that this occurred a few times in my life, 4.0% that it occured a few times per year and 2.0% stated that it occured weekly or daily. Most participants (83.8%) never felt perceived as being dishonest, and 81.8% never felt threatened by others. (Table 3)

Regarding cancer stigma, we found significant gender differences for those who would not feel comfortable around someone with cancer (P < 0.001), concerning perceptions of cancer patients being normal (P < 0.001), the need to prioritize the needs of people with cancer (P < 0.001), perceptions of a cancer diagnosis being the fault of the individual (P < 0.001), that cancer is more frightening than other diseases (P < 0.001), and that women worry about getting cancer (P < 0.001). Among women, 19.0% agreed and 25.0% strongly agreed while among men, 16.1% agreed and only 1.8% strongly agreed that "I would not feel comfortable around someone with cancer." Among women, 31.0% agreed and 57.0% strongly agreed while among men, 41.1% agreed and 8.9% strongly agreed that once you've had cancer you're never 'normal' again. Among women, 31.3% agreed and 32.3% strongly agreed while among men, 3.6% agreed and 5.4% strongly agreed that the health care needs of people with cancer should not be prioritized. Among women, 16.0% agreed and 27.0% strongly agreed while among men, 12.5% agreed and 1.8% strongly agreed that if a person has cancer it's probably their fault. Among women, 34.3% agreed and 50.5% strongly agreed while among men, 36.4% agreed and 16.4% strongly agreed that cancer is more frightening than most other diseases. Among women 48.0% strongly agree while only 7.1% of men strongly agree that other women often state that they are worried about getting cancer. We found no significant difference between women and men in stating that "I would feel sorry for someone with cancer." Among all respondents, 48.1% agree and 40.4% strongly agree. (Table 4)

**Table 3. Adapted everyday discrimination scale by gender.**

| | Female (N = 101) | Male (N = 57) | Total (N = 158) | p value |
|---|---|---|---|---|
| **Feel treated with less courtesy or respect than others** | | | | 0.093 |
| Every day | 9 (9.2%) | 0 (0.0%) | 9 (5.9%) | |
| Every week | 2 (2.0%) | 0 (0.0%) | 2 (1.3%) | |
| A few times per year | 6 (6.1%) | 3 (5.6%) | 9 (5.9%) | |
| A few times in my life | 22 (22.4%) | 10 (18.5%) | 32 (21.1%) | |
| Never | 59 (60.2%) | 41 (75.9%) | 100 (65.8%) | |
| **Feel treated with less courtesy or respect by their spouse** | | | | 0.001 |
| Every day | 14 (14.3%) | 0 (0.0%) | 14 (9.3%) | |
| Every week | 4 (4.1%) | 0 (0.0%) | 4 (2.7%) | |
| A few times per year | 15 (15.3%) | 2 (3.8%) | 17 (11.3%) | |
| A few times in my life | 18 (18.4%) | 9 (17.3%) | 27 (18.0%) | |
| Never | 47 (48.0%) | 41 (78.8%) | 88 (58.7%) | |
| **Feel that others act as if they are not smart** | | | | 0.031 |
| Every day | 1 (1.0%) | 0 (0.0%) | 1 (0.6%) | |
| Every week | 1 (1.0%) | 0 (0.0%) | 1 (0.6%) | |
| A few times per year | 4 (4.0%) | 5 (9.4%) | 9 (5.8%) | |
| A few times in my life | 16 (15.8%) | 17 (32.1%) | 33 (21.4%) | |
| Never | 79 (78.2%) | 31 (58.5%) | 110 (71.4%) | |
| **Feel perceived as being dishonest** | | | | 0.109 |
| Every day | 0 (0.0%) | 0 (0.0%) | 0 (0.0%) | |
| Every week | 0 (0.0%) | 0 (0.0%) | 0 (0.0%) | |
| A few times per year | 1 (1.0%) | 2 (4.2%) | 3 (2.0%) | |
| A few times in my life | 11 (11.0%) | 10 (20.8%) | 21 (14.2%) | |
| Never | 88 (88.0%) | 36 (75.0%) | 124 (83.8%) | |
| **Feel threatened by others** | | | | 0.422 |
| Every day | 1 (1.0%) | 0 (0.0%) | 1 (0.7%) | |
| Every week | 0 (0.0%) | 0 (0.0%) | 0 (0.0%) | |
| A few times per year | 3 (3.0%) | 3 (6.1%) | 6 (4.1%) | |
| A few times in my life | 16 (16.2%) | 4 (8.2%) | 20 (13.5%) | |
| Never | 79 (79.8%) | 42 (85.7%) | 121 (81.8%) | |

We found significant gender differences for certain cancer related attitudes such as the need to get cancer testing or treatment even if it is unpleasant (P = 0.044), the desire for family to know of a personal cancer diagnosis (P < 0.001), and the personal desire to know of a cancer diagnosis in a family member (P < 0.001). Among women, 18.0% disagreed and 10.0% strongly disagreed while among men, 3.6% disagreed and 1.8% strongly disagreed that "If I had cancer, I would want my family to know that I have it." Among women, 25.3% disagreed and 4.0% strongly disagreed while among men, 0% disagreed and 1.8% strongly disagreed that "if someone else in my family had cancer, I would want to know that they have it." Among all respondents a considerable number agreed (50.6%) or strongly agreed (32.5%) that cancer testing or treatment that is unpleasant is worth getting if it would help them to live longer. In addition, among all respondents a considerable number agreed (37.0%) or strongly agreed (57.8%) that if they had cancer, they would want to know that they have it. Among all respondents, 32.1% agreed and 11.5% strongly agreed that getting a serious disease like cancer is fate, there is nothing they can do to change fate. (Table 5)

When making decisions about one's healthcare, women are more likely than men to trust social contacts such as their spouse (46.5% vs 5.3%, p < 0.001), their children (10.9% vs 0%,

**Table 4. Adapted cancer stigma scale by gender.**

| | Female (N = 101) | Male (N = 57) | Total (N = 158) | p value |
|---|---|---|---|---|
| **I would not feel comfortable around someone with cancer.** | | | | < 0.001 |
| Strongly Disagree | 11 (11.0%) | 11 (19.6%) | 22 (14.1%) | |
| Disagree | 44 (44.0%) | 31 (55.4%) | 75 (48.1%) | |
| Undecided | 1 (1.0%) | 4 (7.1%) | 5 (3.2%) | |
| Agree | 19 (19.0%) | 9 (16.1%) | 28 (17.9%) | |
| Strongly Agree | 25 (25.0%) | 1 (1.8%) | 26 (16.7%) | |
| **Once you've had cancer you're never normal again.** | | | | < 0.001 |
| Strongly Disagree | 2 (2.0%) | 7 (12.5%) | 9 (5.8%) | |
| Disagree | 8 (8.0%) | 12 (21.4%) | 20 (12.8%) | |
| Undecided | 2 (2.0%) | 9 (16.1%) | 11 (7.1%) | |
| Agree | 31 (31.0%) | 23 (41.1%) | 54 (34.6%) | |
| Strongly Agree | 57 (57.0%) | 5 (8.9%) | 62 (39.7%) | |
| **The health care needs of people with cancer should not be prioritized.** | | | | < 0.001 |
| Strongly Disagree | 9 (9.1%) | 18 (32.1%) | 27 (17.4%) | |
| Disagree | 24 (24.2%) | 31 (55.4%) | 55 (35.5%) | |
| Undecided | 3 (3.0%) | 2 (3.6%) | 5 (3.2%) | |
| Agree | 31 (31.3%) | 2 (3.6%) | 33 (21.3%) | |
| Strongly Agree | 32 (32.3%) | 3 (5.4%) | 35 (22.6%) | |
| **If a person has cancer it is probably their fault.** | | | | < 0.001 |
| Strongly Disagree | 11 (11.0%) | 11 (19.6%) | 22 (14.1%) | |
| Disagree | 40 (40.0%) | 23 (41.1%) | 63 (40.4%) | |
| Undecided | 6 (6.0%) | 14 (25.0%) | 20 (12.8%) | |
| Agree | 16 (16.0%) | 7 (12.5%) | 23 (14.7%) | |
| Strongly Agree | 27 (27.0%) | 1 (1.8%) | 28 (17.9%) | |
| **I would feel sorry for someone with cancer.** | | | | 0.140 |
| Strongly Disagree | 1 (1.0%) | 4 (7.1%) | 5 (3.2%) | |
| Disagree | 9 (9.0%) | 3 (5.4%) | 12 (7.7%) | |
| Undecided | 0 (0.0%) | 1 (1.8%) | 1 (0.6%) | |
| Agree | 48 (48.0%) | 27 (48.2%) | 75 (48.1%) | |
| Strongly Agree | 42 (42.0%) | 21 (37.5%) | 63 (40.4%) | |
| **I feel that cancer is more frightening than most other diseases.** | | | | < 0.001 |
| Strongly Disagree | 0 (0.0%) | 6 (10.9%) | 6 (3.9%) | |
| Disagree | 11 (11.1%) | 15 (27.3%) | 26 (16.9%) | |
| Undecided | 4 (4.0%) | 5 (9.1%) | 9 (5.8%) | |
| Agree | 34 (34.3%) | 20 (36.4%) | 54 (35.1%) | |
| Strongly Agree | 50 (50.5%) | 9 (16.4%) | 59 (38.3%) | |
| **Other women often state that they are worried about getting cancer.** | | | | < 0.001 |
| Strongly Disagree | 0 (0.0%) | 1 (1.8%) | 1 (0.6%) | |
| Disagree | 3 (3.0%) | 1 (1.8%) | 4 (2.6%) | |
| Undecided | 24 (24.0%) | 30 (53.6%) | 54 (34.6%) | |
| Agree | 25 (25.0%) | 20 (35.7%) | 45 (28.8%) | |
| Strongly Agree | 48 (48.0%) | 4 (7.1%) | 52 (33.3%) | |

p = 0.008), or other family members (17.8% vs 3.5%, p = 0.011). Men however are more likely than women to trust individuals within the health system such as a physician (22.8% vs 2.0%, p < 0.001), a nurse (50.9% vs 18.8%, p < 0.001), or a community health worker (21.1% vs 0.0%, p < 0.001). (See Fig 1) Furthermore, men are more likely than women to say that they

**Table 5. Cancer attitudes by gender.**

| | Female (N = 101) | Male (N = 57) | Total (N = 158) | p value |
|---|---|---|---|---|
| **Cancer testing or treatment that is unpleasant is worth getting if it would help me to live longer** | | | | 0.044 |
| Strongly Disagree | 2 (2.0%) | 1 (1.9%) | 3 (1.9%) | |
| Disagree | 8 (8.0%) | 0 (0.0%) | 8 (5.2%) | |
| Undecided | 9 (9.0%) | 6 (11.1%) | 15 (9.7%) | |
| Agree | 44 (44.0%) | 34 (63.0%) | 78 (50.6%) | |
| Strongly Agree | 37 (37.0%) | 13 (24.1%) | 50 (32.5%) | |
| **If I had cancer, I would want to know that I have it** | | | | 0.110 |
| Strongly Disagree | 1 (1.0%) | 1 (1.8%) | 2 (1.3%) | |
| Disagree | 5 (5.1%) | 0 (0.0%) | 5 (3.2%) | |
| Undecided | 0 (0.0%) | 1 (1.8%) | 1 (0.6%) | |
| Agree | 40 (40.4%) | 17 (30.9%) | 57 (37.0%) | |
| Strongly Agree | 53 (53.5%) | 36 (65.5%) | 89 (57.8%) | |
| **If I had cancer, I would want my family to know that I have it.** | | | | < 0.001 |
| Strongly Disagree | 10 (10.0%) | 1 (1.8%) | 11 (7.1%) | |
| Disagree | 18 (18.0%) | 2 (3.6%) | 20 (12.8%) | |
| Undecided | 3 (3.0%) | 3 (5.4%) | 6 (3.8%) | |
| Agree | 39 (39.0%) | 17 (30.4%) | 56 (35.9%) | |
| Strongly Agree | 30 (30.0%) | 33 (58.9%) | 63 (40.4%) | |
| **If someone else in my family had cancer, I would want to know that they have it.** | | | | < 0.001 |
| Strongly Disagree | 4 (4.0%) | 1 (1.8%) | 5 (3.2%) | |
| Disagree | 25 (25.3%) | 0 (0.0%) | 25 (16.1%) | |
| Undecided | 0 (0.0%) | 0 (0.0%) | 0 (0.0%) | |
| Agree | 38 (38.4%) | 23 (41.1%) | 61 (39.4%) | |
| Strongly Agree | 32 (32.3%) | 32 (57.1%) | 64 (41.3%) | |
| **Getting a serious disease like cancer is fate, there is nothing I can do to change fate** | | | | 0.139 |
| Strongly Disagree | 24 (24.0%) | 11 (19.6%) | 35 (22.4%) | |
| Disagree | 27 (27.0%) | 10 (17.9%) | 37 (23.7%) | |
| Undecided | 6 (6.0%) | 10 (17.9%) | 16 (10.3%) | |
| Agree | 30 (30.0%) | 20 (35.7%) | 50 (32.1%) | |
| Strongly Agree | 13 (13.0%) | 5 (8.9%) | 18 (11.5%) | |

**Fig 1. Trusted opinion for healthcare decisions.**

have the final say at home regarding their own healthcare decisions (78.9% vs 16.0%, p < 0.001), while women are more likely than men to state that their spouse has the final say (72.0% vs 8.8%, p < 0.001). When women were asked, specifically, about opinions regarding their decision to get screened for cervical cancer, 55.4% of women stated that the head of the household would be most influential. (See Fig 2)

Although there were no statistically significant gender differences in questions related to screening recommendations, it is noteworthy that among all respondents, 19.7% strongly agree, 41.4% agree, and 32.5% remain undecided that overall, other women that they know recommend the cervical cancer test. In addition, 49.4% strongly agree, 37.2% agree, and 9.0% remain undecided that they would recommend that women get routine testing for cervical cancer. (Table 6)

## Screened correlation

Subsequently, we correlated all variables with screening behavior among women. S1 Table lists these results. Among women who have been screened once, 92.3% strongly recommend that other women get screened compared to 44.0% of women never screened. (P = 0.006) Among women who have never been screened, 48.2% disagree and 4.7% strongly disagree that they would not feel comfortable around someone with cancer. However, among those who have been screened a single time, 15.4% disagree and 46.2% strongly disagree with this statement (P < 0.001). Among women who were screened one time, 61.5% strongly agreed that if a person has cancer it's probably their fault compared to 22.4% of women who have never been screened (P = 0.003). In addition, 76.9% of women who have been screened one time strongly agreed that other women often state that they are worried about getting cancer compared to only 44.7% of women never screened (P = 0.019).

In correlating cancer attitudes with screening behavior we discovered that 91.7% of women screened a single time strongly agreed that they would want to know if they were diagnosed with cancer compared with 49.4% of women never screened (P = 0.039). However, results indicating desire for a family member to know about diagnosis were insignificant. It is notable that on this question nearly half of those who had been screened one time either disagreed (23.1%) or strongly disagreed (23.1%) that they would want a family member to know about their diagnosis compared to 17.6% and 8.2% respectively for women never screened. We found that among both women who have never been screened and those who have been screened one time, most prefer to know about the diagnosis of someone else in the family. Of those who have never been screened, 40.5% agree and 29.8% strongly agree that they would

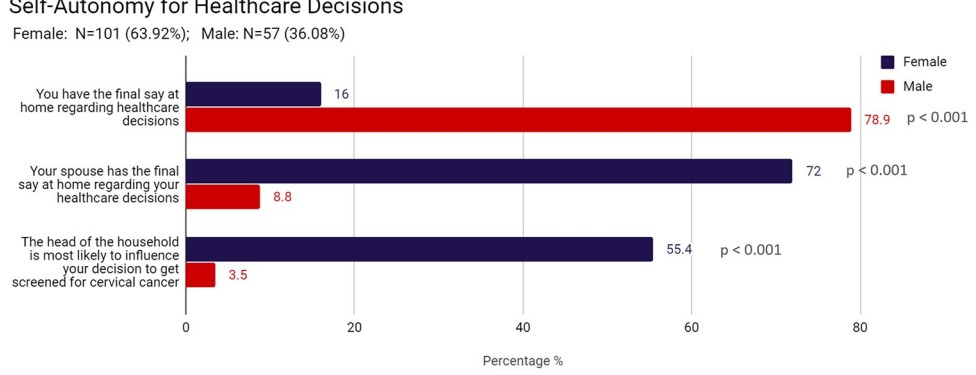

**Fig 2. Self-autonomy for healthcare decisions.**

**Table 6. Cervical cancer screening recommendation by gender.**

| | Female (N = 101) | Male (N = 57) | Total (N = 158) | p value |
|---|---|---|---|---|
| **Other women that I know recommend the cervical cancer test** | | | | 0.832 |
| Strongly Disagree | 2 (2.0%) | 1 (1.8%) | 3 (1.9%) | |
| Disagree | 4 (4.0%) | 3 (5.4%) | 7 (4.5%) | |
| Undecided | 30 (29.7%) | 21 (37.5%) | 51 (32.5%) | |
| Agree | 43 (42.6%) | 22 (39.3%) | 65 (41.4%) | |
| Strongly Agree | 22 (21.8%) | 9 (16.1%) | 31 (19.7%) | |
| **I would recommend that women get routine testing for cervical cancer** | | | | 0.428 |
| Strongly Disagree | 3 (3.0%) | 1 (1.8%) | 4 (2.6%) | |
| Disagree | 3 (3.0%) | 0 (0.0%) | 3 (1.9%) | |
| Undecided | 11 (11.1%) | 3 (5.3%) | 14 (9.0%) | |
| Agree | 33 (33.3%) | 25 (43.9%) | 58 (37.2%) | |
| Strongly Agree | 49 (49.5%) | 28 (49.1%) | 77 (49.4%) | |

want to be informed. Among those screened once, 15.4% agree and 53.9% strongly agree that they would like to be informed of a family member's diagnosis. (P = 0.041) Among women screened a single time, 61.5% strongly disagreed that cancer is fate and there is nothing they can do to change fate compared with 18.8% of never screened women who strongly disagreed with this statement. (P = 0.012) (Table not shown. See S1 Table)

## Education effect on gender perceptions

In examining the effect of education on gender perceptions across all variables, it should be noted that 97% of the women in the study received no more than primary education, while 23.1% of men received some secondary schooling or above. We, therefore, compared the effect of Quranic school or no education to primary education or above across all variables. Through this analysis, we identified statistically significant findings for education level relative to perceptions of a woman's role as caretaker (P < 0.001) and a man's role as decision maker (P < 0.001). Among women with "Higher Education" 47.5% disagreed with the statement "A woman's most important role is to take care of her home and cook for her family" and only 11.7% of women with "Low Education" disagreed while among men, 3.8% and 16.7% of those with "Low Education" and "Higher Education" respectively disagreed. Among women with "Higher Education" 37.5% disagreed with the statement "A man should have the final word about decisions in his home" and only 14.8% of women with "Low Education" disagreed while among men, 10.3% of those with "Higher Education" disagreed. No men classified as having "Low Education" disagreed with this statement. (See S2 Table)

Concerning the adapted everyday discrimination scale analyzed with the gender-education composite variable, we found significant results for a) 'generally feel treated with less courtesy than others' (P = 0.008), b) 'feel treated with less respect by their spouse,' (P = 0.010) c) 'perceived as unintelligent by others' (P < 0.001), d) 'feel perceived as being dishonest' (P = 0.008), and e) 'feel threatened by others' (P = 0.042). (See S3 Table) Regarding the adapted cancer stigma scale, we found significant results regarding a) 'comfort with being around someone with cancer' (P < 0.001), b) 'those diagnosed with cancer are no longer normal' (P < 0.001), c) 'the healthcare needs of people with cancer should not be prioritized' (P < 0.001), d) 'having cancer is probably the fault of that person' (P < 0.001), e) 'cancer is more frightening than other diseases' (P < 0.001), and f) 'other women often state that they are worried about getting cancer' (P < 0.001),. Illustrating this effect of education, we found that 59.0% of women with "Higher Education" disagreed that having cancer is the fault of that person, while 27.9% of

women who had "Low Education" disagreed with this statement. Among men, we found that 50.0% and 34.5% of those with "Low" and "Higher" education respectively strongly disagreed with the statement. (See S4 Table) In addition, we found significant effect of education on gender perceptions on other cancer attitudes including: a) the value of cancer testing, even if unpleasant (P = 0.011), b) desire to be informed about one's personal cancer diagnosis (P = 0.024), c) desire for family to be aware of one's cancer diagnosis (P = 0.002), c) desire to know of cancer diagnosis for another family member (P < 0.001), and d) getting cancer is fate (P = 0.001). (See S5 Table) We did not find any significant impact of education on gender perceptions of recommendation for the cervical cancer screening test. (See S6 Table)

## Discussion

We have identified differences by gender in the perception of gender roles, everyday discrimination, cancer stigma indicators, other cancer-related attitudes, trusted opinion in healthcare decisions, and autonomy in healthcare decision-making. These findings reveal structural gender disadvantages and important insights related to social influences that may play a role in decision-making and screening behavior. Our findings concerning social influences are valuable in illustrating the importance of better understanding key social norms.

### Structural gender disadvantages

Our data reveal some structural disadvantages for women within our study population. Women were less educated than men, and, in turn, were less likely to speak the national languages, Wolof and French. Education and language are both meaningful social determinants of health [38] and may be indicators of status and empowerment, both of which are key to accessing healthcare within this population. In addition, the social role for women in Southeastern Sénégal appears to be largely agreed upon between women and men. In this region, women generally work in the home and men conduct business and make decisions on the part of the family. Somewhat more women than men disagreed with these prescribed roles lending weight to the desire by some women for more autonomy.

Our data also indicate gender differences among variables in the adapted everyday discrimination scale within this population. Women were much more likely to state that they are treated with less courtesy or respect by their spouse on a frequent basis. In contrast, feeling general disrespect outside of the home are not significant. This finding may, therefore, be closely tied to the accepted social roles of women and men. Interestingly, men were somewhat more likely to feel perceived as being unintelligent. We have no explanatory mechanism, but this phenomenon may be related to men being much more likely to take on the role of leader in business interactions. In doing so, men are much more likely to travel and engage with others who have higher levels of education. They, therefore, may be comparing themselves to a different audience than women.

### Attitudes and stigma

Our findings indicate that women are much more likely to personally stigmatize cancer. Women are significantly more likely than men to state that a) they would feel uncomfortable around someone with cancer, b) someone with a cancer diagnosis is never normal again, c) the health needs of people with cancer should not be prioritized, d) if a person has cancer it is probably their fault, e) I feel that cancer is more frightening than most other diseases, and f) other women often state that they are worried about getting cancer. Both women and men agreed on the importance of seeking screening or treatment for cancer, as well as the desire to know personally about a screening result. However, we found a significant difference between

women and men concerning the desire for others to know about a personal cancer diagnosis. We found a similar pattern concerning the desire to know about a family member's cancer diagnosis. This may indicate that women are also more likely to have a perceived stigma against cancer. These findings are intriguing and warrant further investigation. Higher rates of both personal and perceived cancer stigma among women has profound implications for how population and community level communication strategies for cancer prevention and control should be designed. Given that social norms play a critical role in the development of stigmas, we need a better understanding of the negative social influences that shape women's and men's knowledge, attitudes, and beliefs, as well as whether early positive social influences may be impactful in increasing early uptake for cancer prevention services.

## Social influences

Our data show that women are much more likely to rely on the guidance and advice of their spouses or others within their immediate social network while men rely on the recommendations of health professionals when considering health care decisions. The large majority of women and men agree that it is the men who have the healthcare seeking decision making power at home. While the individual variables in our study (categorized as attitudes, discrimination, and stigma) are not altogether specific to cervical cancer, our data does indicate that both women and men would recommend the cervical cancer screening test to others. The perceived recommendation of the screening test from other women, however, is not as considerable.

## Screening behavior

Women who have been screened are much more likely than non-screened women to recommend that other women get routine cervical cancer screening. In addition, it appears that they are less likely to display personal stigma toward cancer patients as evidenced by disagreement with having discomfort around a cancer patient. Screened women do note that other women are worried about getting cancer. If they were personally diagnosed or someone in their family were diagnosed, they would overwhelmingly want to know. However, on the question to inform family members of their own diagnosis, they are split. Further inquiry to explore this contradictory finding could pursue whether some women prefer others not be informed of a diagnosis because of shame, perceived stigma, stoicism, or other potential reasons.

Interestingly, screened women tended to strongly agree that a diagnosis of cancer is the fault of the person. Even though all questions were field tested, this may indicate confusion underlying the premise of the question. It is not possible to know if the women who responded in this way have responded specific to the context of cervical cancer and are indicating that they feel that screening is a way to prevent this illness, and, therefore, it is the responsibility of the individual to seek this service. Lending weight to this supposition is the fact that 61.5% of screened women strongly disagreed that cancer is fate. This significant finding illustrates the importance of ensuring that women feel empowered to prevent cervical cancer through screening by ensuring that stigma, attitudes, and beliefs are prioritized through outreach efforts aimed at uptake.

## Effect of educational achievement on gender perceptions

Our findings exploring the effect of educational achievement on gender perceptions have considerable implications for future work. Concerning a woman's role, there is a clear correlation with increased education among women and men and disagreement with the statement outlining a very traditional perspective. Likewise, increased levels of education indicate that both

men and women are more likely to disagree that men are the sole decision maker in the family. We see similar attenuation of gender specific findings on the adapted everyday discrimination scale, the adapted cancer scale, and other cancer-specific attitudes and opinions. This brings to light critical questions to be addressed through future research, the results of which could impact the development of educational, behavioral change, and social mobilization focused interventions aimed at cancer and, specifically, cervical cancer prevention and control at various levels (individual, household, community, organizational, and policy).

## Confronting barriers and context

Our findings that illustrate that gender roles in decision making, gender influences in discrimination and cancer stigmas, and other structural barriers such as educational attainment and language are meaningful social determinants of health related to cervical cancer screening uptake in Senegal. The existing literature exploring these themes is not robust but presents some insights that may be helpful in further interpreting our findings and guiding our next steps. As an example, cultural norms, gender roles, knowledge, and stigma were identified as socio-cultural factors influencing a woman's decision to seek cervical cancer screening in disadvantaged communities in Cape Town, South Africa. [39] In addition, a study in Cusco, Peru linked underlying determinants such as fear, embarrassment, community conversations about cervical cancer, willingness to talk about cervical cancer, and gender dynamics, including spousal support, to health communications preferences. They found that cultural misconceptions and male perspectives were significant factors predicting screening uptake with an overwhelming need for interventions addressing sociocultural influences in order to address the underlying root causes. [40] These findings suggest that behavior interventions aimed at increasing the uptake of cervical cancer screening services should utilize strategies that go beyond simple health communication and education. In fact, one study examining factors related to breast and cervical cancer screening uptake among Cambodian and Thai women in Southern California identified similar structural barriers that these women were, in large part, unable to overcome such barriers without the assistance of a community navigator. [41] While it is critical to address knowledge gaps, our challenge will be to concurrently address underlying social determinants of cervical cancer screening uptake through positive social influence.

## Limitations

There were some limitations in the methodology used to recruit households within each site. It was assumed that every structure identified on the map was a household and it was thus, included in the count because specific household information was not provided. In reality, some structures were businesses, vacant, or not present. When encountering this scenario, we progressed to the next marked structure on the list. In addition, the satellite images used to create the maps on OpenStreetMaps may have been outdated. In some cases data collection was attempted separately from the recruitment and consent process. This may have resulted in the inability to collect data from some of the selected households due to the participant being away during the day(s) that the research assistant was present. Considerable effort was made to select communities that are representative of the immediate and surrounding districts and regions of rural Sénégal. However, our sample may not be generalizable to other areas of Sénégal or other countries in West Africa. In addition, we must use caution in interpreting these results given the low numbers of women within our study sample who have been screened for cervical cancer. We will follow the trend of these indicators over time as women are exposed to the peer education program.

## Conclusions

The findings concerning social influences are valuable in illustrating the importance of better understanding key social norms and the contexts in which they are found or implicated. Our findings illustrate the critical need, as well, to recognize gender differences concerning social influences within the same context. By detailing the potential negative social influences that can directly act as or contribute to barriers to healthcare services utilization, community outreach activities including social and behavior change communication strategies aimed at these factors can help to overcome existing challenges in cervical cancer prevention and control. Furthermore, the development of innovative interventions such as patient navigation programs that incorporate or leverage positive social influences may prove useful in optimizing health services uptake such as HPV vaccination and cervical cancer screening earlier and more effectively.

Our study also illustrates that gender norms should be routinely considered in efforts aimed at improving uptake of cervical cancer health services in the rural region of Kédougou, Sénégal. In this vein, women's likelihood of being more susceptible to some types of everyday discrimination should guide discrete and sensitive interventions at the health service level as well as within the community setting. Women's greater likelihood of harboring personal stigma and being susceptible to perceptions of perceived stigmas should be openly and proactively addressed through the identification and interruption of negative social influences while positive social influences aimed at overcoming these sensitive challenges should be fully leveraged. Likewise, attitudes and opinions that indicate that men currently maintain a considerable role in the healthcare utilization process should be emphasized in the short term and coupled with parallel activities that seek to empower women in the long-term. Men's role in advancing education and healthy healthcare decision making through positive social influence should be leveraged alongside efforts focused on women.

The differences observed in the analysis exploring the effect of education on gender perceptions illustrate that some gender differences may be attenuated with more knowledge and advancement through formal education. However, it is important to recognize the underlying social fabric in this rural region as the immediate context in seeking the goal of increased health service uptake for cervical cancer screening services here. The reality of this somewhat isolated and underdeveloped region is that social determinants such as gender impact efforts aimed to improve cervical cancer prevention and control in the region. It is critical, of course, to consider a human rights approach and address underlying social determinants through a long-term vision. However, the reality of uniformly advancing education and addressing existing cultural influences that result in social norms that are considered problematic to achieving health equity is daunting. Therefore, efforts to advance the goal of the elimination of cervical cancer should, in the short-term, seek to gain a more profound understanding of the ways that gender, language, and other social determinants impact immediate barriers addressable through interventions. Social and behavior change communication coupled with a community-based patient navigation program may be one approach that can focus both on education while seeking to leverage the social influences that exist in achieving immediate and long-term goals.

## Supporting information

**S1 Data.**
(CSV)

**S1 File.**
(PDF)

**S2 File.**
(RMD)

**S1 Questionnaire.**
(DOCX)

**S2 Questionnaire.**
(DOCX)

**S1 Table.**
(DOC)

**S2 Table.**
(DOC)

**S3 Table.**
(DOC)

**S4 Table.**
(DOC)

**S5 Table.**
(DOC)

**S6 Table.**
(DOC)

## Acknowledgments

The authors would like to acknowledge and thank the following individuals including all officials at the Kedougou regional level including Cheikh Senghor MD and Dr. David Ngom MD. We are also grateful to other health system personnel at the Kedougou district: Fatoumata Traore, Moussa Ndiaye MD, Marguerite Thiare, Bakary Boubou Traore, the Saraya District: Evrard Kabou MD, Daouda Gueye, and the Salemata District: Mamadou Moustapha Thioub. We are also extraordinarily indebted to the local research assistants who ensured that all of the work was accomplished. These amazing individuals include Hawa Diallo, Fatoumata Dia, Dib Faye, Tahibou Niang, Lamine Doucare, Moussa Salife Didibe, and Moussoucouta Samoura. We are also very grateful to the Peace Corps administrators and volunteers who have provided in-kind support for this project over the years. These extraordinary individuals include Chris Hedrick, Cheryl Faye, Mamadou Diaw, Vanessa Dickey, Maureen Cunningham, Adji Thiaw, Imane Sene, Pape Camara, Chris Brown, Leah Moriarty, Meera Sarathy, Marielle Goyette, Larocha LaRiviere, Chip Ko, Ivy Renfro, Annē Linn, Patrick Linn, Katie Wallner, Chris Coox, Sarah Mollenkopf, Aaron Persing, Tess Komarek, Laurie Ohlstein, Emily Johnson, Arielle Kempinsky, Lesa Young, Carmen Dibaya, Maria Castrillon, Ethan Quinn, Gracey McGrory, Aaron Macoubray, Sherry Vazhayil, Ashley Prettyman, Hans-Martin Ishida, Brendan Gray, Elizabeth Costello, Emma Murphy, Cason Kirby, Emma Luu Van Lang, and Gina Siedow. In addition, we are grateful for those who have provided in-kind support from the University of Illinois at Chicago for this project. Many thanks to John Hickner MD, Memoona Hasnain MD MHPE PhD, Stevan Weine MD, Marc Atkins PhD, Michael Berbauma PhD as well as those who have supported the grant management at the Institute of Health Policy and Research including Julieth Pineros, Erika Magallenos, Rocio Bueno, and Przemyslaw Racinski among many others. We are also grateful to David Peters MD DrPH for mentorship in the development of this project.

## Author Contributions

**Conceptualization:** Hamidou Thiam, Zola Collins, Elly Lou De Jesus, Youssoupha Ndiaye, Jennifer S. Smith, Marian Fitzgibbon, Jon Andrew Dykens.

**Data curation:** Jon Andrew Dykens.

**Formal analysis:** Natalia Ongtengco, Tianxiu Wang, Jon Andrew Dykens.

**Funding acquisition:** Jon Andrew Dykens.

**Investigation:** Jon Andrew Dykens.

**Methodology:** Zola Collins, Elly Lou De Jesus, Caryn E. Peterson, Ellen Hendrix, Jon Andrew Dykens.

**Project administration:** Ellen Hendrix, Jon Andrew Dykens.

**Resources:** Jon Andrew Dykens.

**Supervision:** Zola Collins, Elly Lou De Jesus, Caryn E. Peterson, Ellen Hendrix, Youssoupha Ndiaye, Babacar Gueye, Omar Gassama, Abdoul Aziz Kasse, Adama Faye, Jennifer S. Smith, Marian Fitzgibbon.

**Validation:** Zola Collins, Elly Lou De Jesus.

**Writing – original draft:** Natalia Ongtengco, Caryn E. Peterson, Jon Andrew Dykens.

**Writing – review & editing:** Hamidou Thiam, Zola Collins, Elly Lou De Jesus, Ellen Hendrix, Youssoupha Ndiaye, Babacar Gueye, Omar Gassama, Abdoul Aziz Kasse, Adama Faye, Jennifer S. Smith, Marian Fitzgibbon, Jon Andrew Dykens.

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
