## [Decision Letter · Decision Letter 0]

19 Feb 2020

PONE-D-20-01502

Role of gender in perspectives of discrimination, stigma, and attitudes relative to cervical cancer in rural Sénégal

PLOS ONE

Dear Dr Sykens,

Thank you for submitting your manuscript to PLOS ONE. After careful consideration, we feel that it has merit but does not fully meet PLOS ONE’s publication criteria as it currently stands. Therefore, we invite you to submit a revised version of the manuscript that addresses the points raised during the review process.

Please review all of the comments, especially relating to the methods and suggestions to make the paper more concise and generally more readable.

We would appreciate receiving your revised manuscript by March 20, 2020. To enhance the reproducibility of your results, we recommend that if applicable you deposit your laboratory protocols in protocols.io, where a protocol can be assigned its own identifier (DOI) such that it can be cited independently in the future. For instructions see: http://journals.plos.org/plosone/s/submission-guidelines#loc-laboratory-protocols

We look forward to receiving your revised manuscript.

Kind regards,

Mellissa H Withers, PhD, MHS

Academic Editor

PLOS ONE

Journal Requirements:

1. Please include additional information regarding the survey or questionnaire used in the study and ensure that you have provided sufficient details that others could replicate the analyses. For instance, if you developed a questionnaire as part of this study and it is not under a copyright more restrictive than CC-BY, please include a copy, in both the original language and English, as Supporting Information.

2. We note that your report two different date ranges for the present study (either Oct 2018-Jan 2019 or Dec 2018-Feb 2019). Please include in your Methods section the exact date ranges over which you recruited participants to this study.

3. We note you have included a table to which you do not refer in the text of your manuscript. Please ensure that you refer to Table 6 in your text; if accepted, production will need this reference to link the reader to the Table.

Reviewers' comments:

Reviewer's Responses to Questions

**Comments to the Author**

1. Is the manuscript technically sound, and do the data support the conclusions?

Reviewer #1: Yes

Reviewer #2: Yes

2. Has the statistical analysis been performed appropriately and rigorously? 

Reviewer #1: Yes

Reviewer #2: Yes

3. Have the authors made all data underlying the findings in their manuscript fully available?

Reviewer #1: Yes

Reviewer #2: Yes

4. Is the manuscript presented in an intelligible fashion and written in standard English?

Reviewer #1: Yes

Reviewer #2: Yes

5. Review Comments to the Author

Reviewer #1: Abstract

With some copy-editing and focus on the key points, the abstract can be considerably shortened. An example is the sentence: “Furthermore, men are more likely than women to say that they have the final say at home….while women are more likely than men to state that their spouse has the final say..” This can be reworked into something more straightforward, such as “Men and women were both more likely to state that men have the final decision around the home..”

Introduction

Second sentence: what does fatal mean in this case? Usually, cancer statistics are given in terms of 5 or 10 year survival or fatality.

Can you provide the reader with more of a sense of screening availability In Senegal? Although gendered attitudes likely do play a big role, if screening is not available or accessible, that will likely play the biggest role in limiting coverage.

Methods

How was your sampling plan and methodology determined? Did you calculate a sample size? Given the small number of women screening, would you have enough of a sample size to determine differences associated with screening?

Results

First paragraph: In the sentence about married, what does “51.5% of participants were in a polygamous household as related to significant gender differences in marital status” mean?

Much of the data presented in tables is repeated in the text. It would make the paper more readable and allow the authors to highlight the key findings if they referred readers to the table, rather than repeating all of the information in both places.

In the “Education Effect on Gender Perceptions”, third sentence, can you clarify what you mean by “we identified statistically significant findings for a woman’s role as caretaker (p<0.001), etc.

Much of the subsequent paragraph in this section is a list of significant findings—I think these should all go into a table with a summary or highlight of important points in the text.

Discussion

Generally, the discussion should start with a summary paragraph of the main findings of your study, followed by a deeper dive into how specific results fit into the overall scientific field, and why they may or may not support existing evidence. This discussion section starts with a repeat of the specific findings, which lacks a summative component. I think the paper would be strengthened with a section that brings together the main findings with some synthesis.

Reviewer #2: Overall comments:

- The subject matter of this paper is very timely and a necessary contribution to the literature on cervical cancer prevention in LMICs.

- The paper was very well written and organized with relevant and up-to-date citations.

INTRODUCTION:

1) There is one sentence at the end of the first paragraph that could use some tweaking for clarity:

"Cervical cancer is an important indicator of larger health system problems, including poor access and the lack of culturally competent communication..."

I understand the point the authors are trying to make here but the way it reads could potentially be misconstrued to those that are not as well versed in the literature. Perhaps including words like "The high incidence and mortality of cervical cancer is an important indicator...

Additionally, it would be helpful to add poor access "to care and screening" to identify the specific access issues in the context of cervical cancer disparities.

METHODS:

1) Do the authors have the numbers for the response rate? It would be good for other researchers attempting this sampling strategy to be aware of the realities of obtaining survey responses through this approach and ways to improved if necessary.

2) Data analysis - did the authors give any thought to looking at regional differences (i.e. urban vs rural) or other breakdowns of the regions with gender differences and cancer attitudes? Sometimes attitudes may vary by urban and rural communities particularly when access to screening is more accessible in urban areas.

RESULTS:

1) The sample size is small for recruitment in 9 different districts. Is there a breakdown of respondents by area or do we assume that the language stratification implies area differences? Some clarification here would be helpful.

DISCUSSION:

1) Screening behavior - the 2nd paragraph of this section gives some discussion on the percentage of women that believe cancer was personal fault and the authors suggest that there may be some issues with how the question was interpreted. Was the question field tested prior to the start of the study? Could it be that the cancer stigma questions did not specify a specific cancer and therefore were answered in the context of cervical cancer only?

2) Last sentence, " ..."prevent cervical cancer through EARLY screening..." perhaps it would be more straightforward to omit the word "early" since ongoing screening is necessary throughout a woman's lifetime.

3) It is not clear what the authors mean by ensuring that attitudes and beliefs are prioritized. In other words, what is the situation in which attitudes and beliefs (positive or negative) are prioritized for cervical cancer screening?

4) It is also important to include and address cancer stigma, especially for cervical cancer and HPV.

5) I did not see where discussion of the findings from this study were linked back to existing literature to draw similarities or differences with other similar works.

CONCLUSION:

1) Sound conclusions and thoughtful approaches for next steps.

6. PLOS authors have the option to publish the peer review history of their article (what does this mean?). If published, this will include your full peer review and any attached files.

Reviewer #1: No

Reviewer #2: No

---

## [Author Response · Author response to Decision Letter 0]

24 Mar 2020

We are extraordinarily grateful for the reviewer and editorial effort taken to provide us with these very thoughtful comments and suggestions on ways to improve this manuscript. We have taken the time to appropriately revise our manuscript and reply in line to reviewer comments below. 

1. Please include additional information regarding the survey or questionnaire used in the study and ensure that you have provided sufficient details that others could replicate the analyses. For instance, if you developed a questionnaire as part of this study and it is not under a copyright more restrictive than CC-BY, please include a copy, in both the original language and English, as Supporting Information.

Response: We have included the French and English copies of the questionnaire as Supporting Information.

2. We note that your report two different date ranges for the present study (either Oct 2018-Jan 2019 or Dec 2018-Feb 2019). Please include in your Methods section the exact date ranges over which you recruited participants to this study.

Response: Thank you for catching this error. We have specified that the date range for data collection was from October 2018 – February 2019. This has been specified in the Methods section and corrected in the abstract and results.

3. We note you have included a table to which you do not refer in the text of your manuscript. Please ensure that you refer to Table 6 in your text; if accepted, production will need this reference to link the reader to the Table.

Response: The reference to Table 6 has been included in the text.

Response: This task has been completed.

Reviewer Comments: 

Reviewer #1: Abstract

R1.01 With some copy-editing and focus on the key points, the abstract can be considerably shortened. An example is the sentence: “Furthermore, men are more likely than women to say that they have the final say at home….while women are more likely than men to state that their spouse has the final say..” This can be reworked into something more straightforward, such as “Men and women were both more likely to state that men have the final decision around the home..”

Response: We have incorporated this editorial suggestion. The sentence now reads:

“Furthermore, men and women were both more likely to state that men have the final decision the healthcare decisions of women (p < 0.001).”

Introduction

R1.02 Second sentence: what does fatal mean in this case? Usually, cancer statistics are given in terms of 5 or 10 year survival or fatality.

Response: We have revised this sentence with the inclusion of the following… “with 1,367 deaths resulting in a age-standardized mortality rate of 29.1 compared to 6.9 globally.”

R1.03 Can you provide the reader with more of a sense of screening availability In Senegal? Although gendered attitudes likely do play a big role, if screening is not available or accessible, that will likely play the biggest role in limiting coverage.

Response: We have clarified that cervical cancer screening remains inaccessible throughout much of rural Senegal but has been available in the region of the study since 2014. We added a citation to reinforce this. The added sentence reads as:

“Cervical cancer screening remains unavailable in many rural areas of Senegal but has been accessible throughout the Kédougou region of Sénégal since 2014 through the efforts of an ongoing partnership.[7]”

Methods

R1.04 How was your sampling plan and methodology determined? Did you calculate a sample size? Given the small number of women screening, would you have enough of a sample size to determine differences associated with screening? 

Response: The target sample size of 225 (135 women and 90 men) was determined based on the need to have a heterogeneous sample across language and district. Our sample was not adequately powered to detect differences between the two groups (screened and unscreened women).

Results

R1.05 First paragraph: In the sentence about married, what does “51.5% of participants were in a polygamous household as related to significant gender differences in marital status” mean?

Response: Thank you for identifying this poorly worded phrase. We have truncated the phrase to create a clear sentence. “…51.5% of participants were in a polygamous household”

R1.06 Much of the data presented in tables is repeated in the text. It would make the paper more readable and allow the authors to highlight the key findings if they referred readers to the table, rather than repeating all of the information in both places.

Response: The authors chose to highlight the significant findings within the text. While this does result in some repetition of information between the tables and the text, our opinion is that it aids the reader in more readily interpreting the essential findings by walking her or him through data. By identifying the significant findings it draws the reader’s attention to the key elements that are further contextualized within the discussion and conclusion. 

R1.07 In the “Education Effect on Gender Perceptions”, third sentence, can you clarify what you mean by “we identified statistically significant findings for a woman’s role as caretaker (p<0.001), etc.

Response: Thank you for bringing attention to this unclear statement. We have reworded this as… “Through this analysis, we identified statistically significant findings for education level relative to perceptions of a woman’s role as caretaker…”

R1.08 Much of the subsequent paragraph in this section is a list of significant findings—I think these should all go into a table with a summary or highlight of important points in the text.

Response: We have clarified that these findings are reported in a table provided in Supplement 2.

Discussion

R1.09 Generally, the discussion should start with a summary paragraph of the main findings of your study, followed by a deeper dive into how specific results fit into the overall scientific field, and why they may or may not support existing evidence. This discussion section starts with a repeat of the specific findings, which lacks a summative component. I think the paper would be strengthened with a section that brings together the main findings with some synthesis.

Response: We have strengthened the Discussion section with an initial brief summary of key findings with a transition into the following paragraphs to provide more detailed interpretation. 

Reviewer #2: 

- The subject matter of this paper is very timely and a necessary contribution to the literature on cervical cancer prevention in LMICs.

- The paper was very well written and organized with relevant and up-to-date citations.

INTRODUCTION:

R2.01 There is one sentence at the end of the first paragraph that could use some tweaking for clarity:

"Cervical cancer is an important indicator of larger health system problems, including poor access and the lack of culturally competent communication..."

I understand the point the authors are trying to make here but the way it reads could potentially be misconstrued to those that are not as well versed in the literature. Perhaps including words like "The high incidence and mortality of cervical cancer is an important indicator...

Additionally, it would be helpful to add poor access "to care and screening" to identify the specific access issues in the context of cervical cancer disparities.

Response: Thank you for this suggestion. These changes have been made. 

METHODS:

R2.02 Do the authors have the numbers for the response rate? It would be good for other researchers attempting this sampling strategy to be aware of the realities of obtaining survey responses through this approach and ways to improve if necessary.

Response: We have clarified the response rate with the following statement… “Our enrollment goal was 225 (135 women and 90 men). With 158 participants (101 women and 57 men), we achieved a functional response rate of 70.2%. The response rate for women was appreciably higher than for men (74.8% and 63.3%, respectively).” 

R2.03 Data analysis - did the authors give any thought to looking at regional differences (i.e. urban vs rural) or other breakdowns of the regions with gender differences and cancer attitudes? Sometimes attitudes may vary by urban and rural communities particularly when access to screening is more accessible in urban areas.

Response: For this initial analysis, we did not explore regional differences. However, these baseline data will be combined with longitudinal data upon completion of a trial evaluating a peer education intervention. At that time, we plan to complete a multi-level evaluation which will take this consideration into account. 

RESULTS:

R2.04 The sample size is small for recruitment in 9 different districts. Is there a breakdown of respondents by area or do we assume that the language stratification implies area differences? Some clarification here would be helpful.

Response: Thank you for this suggestion. We recruited participants across three sites from each of three districts. We have now revised Table 1 to report the participation across these sites. 

DISCUSSION:

R2.05 Screening behavior - the 2nd paragraph of this section gives some discussion on the percentage of women that believe cancer was personal fault and the authors suggest that there may be some issues with how the question was interpreted. Was the question field tested prior to the start of the study? Could it be that the cancer stigma questions did not specify a specific cancer and therefore were answered in the context of cervical cancer only?

Response: Thank you for raising this point. We have edited the text to incorporate these points. The text now reads…

“Interestingly, screened women tended to strongly agree that a diagnosis of cancer is the fault of the person. Even though all questions were field tested, this may indicate confusion underlying the premise of the question. It is not possible to know if the women who responded in this way have responded specific to the context of cervical cancer and are indicating that they feel that screening is a way to prevent this illness, and, therefore, it is the responsibility of the individual to seek this service.”

R2.06 Last sentence, " ..."prevent cervical cancer through EARLY screening..." perhaps it would be more straightforward to omit the word "early" since ongoing screening is necessary throughout a woman's lifetime.

It is not clear what the authors mean by ensuring that attitudes and beliefs are prioritized. In other words, what is the situation in which attitudes and beliefs (positive or negative) are prioritized for cervical cancer screening?

It is also important to include and address cancer stigma, especially for cervical cancer and HPV.

Response: These are appreciated points. We have revised accordingly. The sentence now reads: 

“This significant finding illustrates the importance of ensuring that women feel empowered to prevent cervical cancer through screening by ensuring that stigma, attitudes, and beliefs are prioritized through outreach efforts aimed at uptake.”

R2.07 I did not see where discussion of the findings from this study were linked back to existing literature to draw similarities or differences with other similar works.

Response: We have added a paragraph at the end of the discussion section that links our findings back to the existing literature. This has strengthened our manuscript as it now better introduces the major themes covered in the conclusion. Thank you for this suggestion. 

CONCLUSION:

- Sound conclusions and thoughtful approaches for next steps.

The contact information for the corresponding author is as follows:

J Andrew Dykens

jdykens@uic.edu

The authors are very grateful to the reviewers for their thoughtful feedback. We believe that the manuscript is much stronger after these revisions. Many thanks for your guidance and consideration.

Thank you for receiving our manuscript and considering it for publication. We appreciate your time and look forward to your response. 

Sincerely,

Andrew Dykens, MD, MPH

Associate Professor of Family Medicine, University of Illinois at Chicago 

Director, Global Health Systems, UIC Center for Global Health

---

## [Decision Letter · Decision Letter 1]

13 Apr 2020

Role of gender in perspectives of discrimination, stigma, and attitudes relative to cervical cancer in rural Sénégal

PONE-D-20-01502R1

Dear Dr. Dykens,

We are pleased to inform you that your manuscript has been judged scientifically suitable for publication and will be formally accepted for publication once it complies with all outstanding technical requirements.

With kind regards,

Mellissa H Withers, PhD, MHS

Academic Editor

PLOS ONE

Additional Editor Comments (optional):

Reviewers' comments:

Reviewer's Responses to Questions

**Comments to the Author**

1. If the authors have adequately addressed your comments raised in a previous round of review and you feel that this manuscript is now acceptable for publication, you may indicate that here to bypass the “Comments to the Author” section, enter your conflict of interest statement in the “Confidential to Editor” section, and submit your "Accept" recommendation.

Reviewer #2: All comments have been addressed

2. Is the manuscript technically sound, and do the data support the conclusions?

Reviewer #2: Yes

3. Has the statistical analysis been performed appropriately and rigorously? 

Reviewer #2: Yes

4. Have the authors made all data underlying the findings in their manuscript fully available?

Reviewer #2: Yes

5. Is the manuscript presented in an intelligible fashion and written in standard English?

Reviewer #2: Yes

6. Review Comments to the Author

Reviewer #2: (No Response)

7. PLOS authors have the option to publish the peer review history of their article (what does this mean?). If published, this will include your full peer review and any attached files.

Reviewer #2: No

---

## [Editor Report · Acceptance letter]

17 Apr 2020

PONE-D-20-01502R1 

Role of gender in perspectives of discrimination, stigma, and attitudes relative to cervical cancer in rural Sénégal 

Dear Dr. Dykens:

I am pleased to inform you that your manuscript has been deemed suitable for publication in PLOS ONE. Congratulations! Your manuscript is now with our production department. 

With kind regards,

on behalf of

Dr. Mellissa H Withers 

Academic Editor

PLOS ONE